# The Road Ahead in Pancreatic Cancer: Emerging Trends and Therapeutic Prospects

**DOI:** 10.3390/biomedicines12091979

**Published:** 2024-09-02

**Authors:** Chris T. P. Do, Jack Y. Prochnau, Angel Dominguez, Pei Wang, Manjeet K. Rao

**Affiliations:** 1Greehey Children’s Cancer Research Institute, University of Texas Health Science Center at San Antonio, San Antonio, TX 78229, USA; dot@uthscsa.edu (C.T.P.D.); prochnau@uthscsa.edu (J.Y.P.); 2Department of Cell Systems and Anatomy, University of Texas Health Science Center at San Antonio, San Antonio, TX 78229, USA; domingueza6@livemail.uthscsa.edu (A.D.); wangp3@uthscsa.edu (P.W.)

**Keywords:** pancreatic cancer, cancer trials, immunotherapy, vaccine

## Abstract

This review explores the challenges and emerging trends in pancreatic cancer therapy. In particular, we focus on the tumor microenvironment and the potential of immunotherapy for pancreatic cancer. Pancreatic ductal adenocarcinoma, characterized by its dense stromal architecture, presents unique challenges for effective treatment. Recent advancements have emphasized the role of the tumor microenvironment in therapeutic resistance and disease progression. We discuss novel strategies targeting the desmoplastic barrier and immunosuppressive cells to enhance immune cell infiltration and activation. Recent clinical trials, particularly those involving novel immunotherapeutic agents and tumor vaccines, are examined to understand their efficacy and limitations. Our analysis reveals that combining immunotherapy with chemotherapy, radiation therapy, or drugs targeting epigenetic processes shows promise, improving overall survival rates and response to treatment. For instance, trials utilizing checkpoint inhibitors in combination with standard chemotherapies have extended disease-free survival by up to 6 months compared to chemotherapy alone. Importantly, vaccines targeting specific tumor neoantigens have shown the potential to increase patient survival. However, these approaches also face significant challenges, including overcoming the immunosuppressive tumor microenvironment and enhancing the delivery and efficacy of therapeutic agents. By providing an overview of both the promising results and the obstacles encountered, this review aims to highlight ongoing efforts to refine immunotherapy approaches for better patient outcomes.

## 1. Introduction

Pancreatic cancer, currently ranked as the tenth most common cancer in the United States, is predicted to become the second leading cause of cancer death by 2030 [1]. Nearly all pancreatic cancer originates in the exocrine cells rather than the endocrine cells (Figure 1A). Furthermore, there are established genetic and modifiable risk factors that significantly increase the likelihood of developing pancreatic cancer. Genetic predispositions, such as mutations in *KRAS* and *CDKN2A*, alongside modifiable factors such as alcohol consumption, chronic pancreatitis, and obesity, play a crucial role in the pathogenesis of the disease (Figure 1B). Understanding these risk factors is essential for early detection and targeted prevention strategies. Five precursor lesions of pancreatic ductal adenocarcinoma (PDAC) have been identified, with the most common progression of the disease involving a transition from acinar cell alterations to ductal metaplasia, advancing to pancreatic intraepithelial neoplasia (PanIN) and culminating in PDAC (Figure 1C) [2,3]. Currently, the gold standard treatment includes surgical resection, if deemed a candidate, alongside radiotherapy and chemotherapy such as gemcitabine or FOLFIRINOX, a combination of chemotherapy drugs: fluorouracil, irinotecan, and oxaliplatin (Figure 1D). However, treatment efficacy is hindered by therapy-associated toxicities and the development of resistance to chemotherapy [4,5]. The indolent nature of pancreatic cancer, alongside its anatomical location and presenting signs, contributes to its diagnosis at metastatic late stages, as outlined by the American Joint Committee on Cancer, with nearly half of all diagnoses presenting at stage 4 [6]. Moreover, the complex tumor microenvironment characteristic of PDAC makes treatment responses unpredictable and complicates the development of new therapies [7]. This late-stage diagnosis, when the disease is already advanced and treatment options are limited, explains the very low five-year survival rate of 13% in PDAC patients [8]. Early diagnosis of PDAC remains a significant challenge because the disease often presents with vague, non-specific symptoms like abdominal pain and weight loss, leading to delayed detection [9]. The pancreas’ deep location complicates early tumor detection through physical exams or imaging. Additionally, there are currently no reliable screening methods for the early detection of PDAC, and as a result, the cancer is often metastatic by the time clinical symptoms appear. While traditional biomarkers like CA19-9 and CEA are often inadequate due to their limited sensitivity and specificity, machine learning algorithms, novel biomarkers, and quantitative assessments of microvesicle-derived DNA and RNA hold promise for early detection [10]. Detecting PDAC early through such combined methods could allow for successful surgical resection, improve survival rates, and enable more targeted therapies. This could potentially enhance the effectiveness of treatment modalities such as immunotherapy, which are less effective in advanced stages.

As the field of targeted therapy continues to evolve, there is an increased focus on a new generation of treatments, with immunotherapy playing a significant role [11]. Agents such as immune checkpoint inhibitors (ICIs) have shown remarkable efficacy in tumors with a high mutation burden [12]. Yet, the response rate of immunotherapy in PDAC patients remains disappointingly low. This issue is in part due to the dense stromal architecture of the disease, which acts as a physical and functional barrier to immune cell infiltration and activity, and partly because the tumor microenvironment (TME) is enriched with immunosuppressive cells [13,14,15].

These issues necessitate a deeper understanding of the PDAC TME to enhance the delivery and efficacy of immunotherapies. In this review, we examine the emerging body of immunotherapeutic interventions for PDAC, focusing on recent clinical trials posted by the National Library of Medicine on clinicaltrials.gov (Table 1). We analyze the shortcomings of recent studies and offer insights and potential directions to refine immunotherapy strategies for PDAC.

## 2. Tumor Microenvironment of PDAC

Understanding the tumor microenvironment (TME) and identifying tumor-intrinsic and -extrinsic factors are critical for advancing immunotherapy in PDAC. The TME, consisting of cancer cells, stromal cells, immune cells, and the extracellular matrix, plays a pivotal role in tumor progression and therapeutic response [16,17]. Tumor-intrinsic factors, such as genetic mutations and aberrant signaling pathways, drive cancer proliferation and resistance to therapy. Conversely, tumor-extrinsic factors, including immune cell infiltration, cytokine milieu, and angiogenesis, influence the efficacy of immunotherapies. In particular, the PDAC TME is marked by a dense population of immunosuppressive cells, such as regulatory T cells (Tregs), myeloid-derived suppressor cells, and tumor-associated macrophages (TAMs) [18]. These cells release inhibitory cytokines, like IL-10 and TGF-β, that dampen the anti-tumor immune response [19]. Additionally, PDAC often presents fewer tumor-specific mutations compared to cancers like melanoma [20]. This results in fewer neoantigens that the immune system can recognize as “foreign”, making it difficult for immune cells to target [21,22]. In addition, tumors often upregulate immune checkpoint proteins, such as PD-L1, on their surfaces [23,24]. When these proteins bind to receptors on immune cells (like PD-1 on T cells), they send an inhibitory signal, preventing the immune cell from attacking the tumor.

PDAC tumors can also alter the metabolic landscape of the TME, creating unfavorable conditions for immune cell function. For instance, they can induce hypoxia or increase the production of immunosuppressive metabolites like adenosine [25,26,27]. PDAC tumors can also reduce the expression of major histocompatibility complex (MHC) class I molecules [28,29]. These molecules are essential for presenting tumor antigens to immune cells. Without them, T cells cannot recognize and target tumor cells effectively [30]. Some PDAC tumors can also release tumor-intrinsic factors that induce apoptosis in effector immune cells, further reducing the body’s ability to mount a defense against the tumor [31,32]. Together, these features create a PDAC TME that is mostly immunosuppressive and makes immunotherapy less effective. By understanding these components, we can develop strategies to modulate the TME, overcoming barriers to immune cell infiltration and activation, thereby enhancing the efficacy of immunotherapeutic approaches in PDAC.

## 3. Role of the Extracellular Matrix and Stiffness in Tumor Invasion

One of the critical features of the PDAC TME is the significant alteration in the extracellular matrix (ECM), particularly its stiffness. PDAC is characterized by a dense, fibrotic stroma, often called desmoplasia, composed of ECM proteins like collagen, fibronectin, and hyaluronan. This desmoplastic reaction not only provides structural support to the tumor but also contributes to the mechanical stiffness of the microenvironment. The increased stiffness of the ECM in PDAC is not merely a byproduct of tumor growth but actively influences tumor behavior. For example, the stiffened ECM enhances mechanotransduction, where cells convert mechanical stimuli into biochemical signals. This activation of mechanotransduction pathways, including PI3K, RhoA-ROCK, YAP/TAZ, and focal adhesion kinase (FAK) signaling, promotes tumor cell survival, proliferation, and invasion [33]. In addition, the stiff ECMs facilitate the migration of tumor cells by creating tracks within the matrix through which cancer cells can invade surrounding tissues [34]. This invasive behavior is further amplified by the ECM’s role in providing resistance to apoptosis and promoting epithelial–mesenchymal transition (EMT), processes crucial for metastatic dissemination [35,36]. The altered ECM and its associated stiffness can also exacerbate hypoxia within the tumor, leading to metabolic reprogramming that supports tumor survival. Lastly, the dense and stiff ECM can physically obstruct the infiltration of immune cells into the tumor core, consequently reducing the effectiveness of immune surveillance and immunotherapies. Moreover, the rigidity of the ECM can alter the phenotype of infiltrating immune cells, often pushing them toward a more immunosuppressive state.

## 4. Targeting the Desmoplastic Barrier in PDAC

The dense desmoplastic stroma in PDAC fosters an immunosuppressive tumor microenvironment. Therefore, targeting stromal components, such as cancer-associated fibroblasts (CAFs) and the signaling pathways facilitating stromal proliferation, offers a promising strategy to mitigate this obstruction and enhance immune cell infiltration and activation. Claudins are key constituents of tight junctions, which are critical structures that regulate the passage of molecules between cells [37,38]. As such, they preserve tissue integrity by establishing barriers between neighboring cells [39]. Two isoforms of CLDN18, CLDN18.2 and CLDN18.1, have distinct localization patterns in specific tissue types [40]. CLDN18.2 is the dominant isoform in normal gastric tissue and is often retained during malignant transformation [41]. This isoform is highly expressed in PDAC, whereas its expression in healthy tissues is limited to the stomach mucosa’s differentiated epithelial cells. It is presumed to be a potential biomarker and target for immunotherapy in PDAC [42]. A recent open-label, multi-site, phase I/IIa clinical trial [NCT04683939] is investigating the efficacy of targeting claudin 18.2 (CLDN18.2) using BNT141, a lipid nanoparticle-encapsulated RNA-based therapy containing two pseudouridine-modified mRNAs that code for monoclonal IgG antibodies against CLDN18.2 [43,44,45] is currently recruiting. The antibody generated by BNT141 is identical in sequence to zolbetuximab (IMAB362), a CLDN18.2-targeted antibody that has demonstrated benefit as an adjunct to chemotherapy in gastric cancers [45]. The fundamental concept of the trial is that anti-CLDN18.2 antibodies encoded by the BNT141 mRNA circulate within the body and selectively bind to cancer cells that exhibit high levels of CLDN18.2 expression and induce tumor cell death predominantly through mechanisms such as antibody-dependent cellular cytotoxicity (ADCC) and complement-dependent cytotoxicity (CDC). A challenge inherent in this therapy is that single-stranded RNA entering dendritic cells may bind to toll-like receptor 7 (TLR7), triggering the release of type I interferons [46,47]. These interferons may promote cancer immune subversion by driving the body toward an immunosuppressive state. This occurs by stimulating increased expression of indoleamine 2,3-dioxygenase (IDO) and interleukin-10 (IL-10), among other anti-inflammatory mediators, promoting a tumor microenvironment conducive to cancer survival [48,49]. In addition to fostering an immune subversive TME, the BNT141 antibody can have toxic side effects due to binding to healthy gastric epithelia. The resultant loss of epithelia may lead to gastrointestinal side effects. Indeed, a multicenter clinical trial reported that up to 6% of participants experienced treatment-related mild adverse events (mainly gastrointestinal toxicities) [50].

## 5. Neutralizing Immunosuppressive Cells

Targeting immunosuppressive cells like Tregs, myeloid-derived suppressor cells (MDSCs), and TAMs offers a pathway to restore anti-tumor immunity [51,52,53]. Exploiting this approach can potentially unlock significant clinical benefits for PDAC patients [54]. For example, agents that deplete MDSCs or inhibit their function can shift the balance toward an immune-promoting environment [55,56,57]. For example, TAMs are known to express leukemia inhibitory factor (LIF), which promotes tumor progression, inflammation, and therapeutic resistance by creating an immunosuppressive TME by inducing the differentiation of Tregs and enhancing the production of other immunosuppressive cytokines [58,59,60]. AZD0171 (also known as MSC-1), a humanized monoclonal antibody targeting LIF with high affinity, induces phenotypic and functional changes in TAMs [61]. These changes promote anti-tumor inflammation and inhibit the proliferation and metabolism of PDAC stem cells, leading to a reduction in tumor growth. This dual targeting, along with new LIF-targeting therapeutic strategies, is currently being evaluated in a clinical trial [NCT04999969]. This trial aims to explore the safety, pharmacokinetics, and overall efficacy of AZD0171 in combination with durvalumab and chemotherapies (gemcitabine and nab-paclitaxel) in patients with metastatic solid cancers. A potential clinical side effect of AZD0171 treatment could be due to its targeting of LIF, which can suppress hematopoietic stem cells. These stem cells depend on LIF for their proliferation, and interference with this pathway may adversely affect their function [62].

## 6. Overcoming Metabolic Barriers

Another strategy to improve immunotherapy response in PDAC patients could be to target metabolic byproducts, as several metabolic byproducts induce immune cell exhaustion and foster a pro-tumor environment [63,64,65]. One such byproduct, CD73, catalyzes the conversion of extracellular ATP/ADP into free adenosine, which then binds to the A_2A_ and A_2B_ receptors, promoting immunosuppression [66,67]. Inhibition of CD73 or targeting A_2A_/A_2B_ receptors to block adenosine receptors may counteract immunosuppression and mitigate the establishment of an immunologically “cold” TME. Efforts to target these pathways are currently underway in multiple clinical trials. One such approach involves the small molecule inhibitor INCB106385, a potent antagonist of A_2A_/A_2B,_ currently under investigation as a monotherapy or in combination with retifanlimab (INCMGA00012), a humanized monoclonal antibody targeting (PD-1) [NCT04580485] [68,69]. As an eligibility criterion, the TME of participants in this clinical trial must have CD8^+^ infiltrating T cells.

While targeting A_2A_/A_2B_ can prevent adenosine-mediated immunosuppression, CD73 is mainly responsible for the production of free extracellular adenosine [70]. Therefore, clinical trials targeting both A_2A_/A_2B_ and CD73 [NCT04989387] could be effective. The drug used to target CD73, INCA00186, is a monoclonal antibody that binds CD73 and inhibits the production of free adenosine [71]. A potential challenge in targeting the adenosine production pathway is side effects such as decreased vasodilation, which could impair lymphocyte migration and infiltration into the TME, and tachycardia, as adenosine slows the heart rate [72,73]. Retifanlimab may counteract these effects directly by alleviating lymphocyte exhaustion through cell-to-cell interaction or/and indirectly by inhibiting the synthesis of adenosine and blocking the receptors that mediate immunosuppression. The dual targeting of CD73 and PD-1 plus established chemotherapeutic regimens like mFOLFIRINOX is currently being evaluated in a clinical trial [NCT05688215] for PDAC. Though CD73 blockade increases the efficacy of PD-1 blockade and reduces levels of IL-1β in the TME, the challenge remains that antibody-based immunotherapeutics such as retifanlimab must first pass through the desmoplastic stroma to reach their targets [74].

Several cytokines, including the immunosuppressive cytokine TGFβ, can collaborate with adenosine to further enhance the immunosuppression [75]. Therefore, targeting TGFβ with CD73 could significantly improve treatment outcomes. Based on this premise, a phase II trial [NCT05632328] that employs AGEN1423, a humanized bifunctional antibody targeting both CD73 and TGFβ, alongside balstilimab, a PD-1 inhibitor, is currently ongoing [76]. This dual-targeting system allows the TME to be transformed from an immunologically “cold” state to a more active, immunologically “hot” condition. One potential caveat of this trial is that treatment with AGEN1423 is associated with increased soluble CD73 (sCD73) levels. This may cause a systemic reduction in tissue-resident memory T cells, which are crucial for long-term immune protection against recurrent infections and cancer surveillance.

## 7. Blocking Immune Checkpoints

Immune checkpoint proteins depend on cell-to-cell interactions that inhibit the effector functions of immune cells [77,78]. The TME often upregulates immune checkpoint ligands like PD-L1, inhibiting T cell functions [23,79]. Checkpoint inhibitors that block PD-1/PD-L1 or CTLA-4 interactions can reinvigorate exhausted T cells and allow for tumor recognition, restoring their cytotoxic functions [80,81,82,83]. Utilizing antibodies that block this inhibitory signal has shown encouraging results in various cancers [84,85,86]. For example, PD-1 antibodies have been investigated as a monotherapy or in combination with various regimens in PDAC patients [87,88]. In one phase II clinical trial [NCT05562297], PD-1 blockade through sintilimab, a human IgG monoclonal antibody against PD-1, is combined with the DNA-damaging agent gemcitabine alongside nab-paclitaxel [89]. As blocking PD-1 can prevent immune cell inhibition, combining such therapy with a DNA-damaging agent such as gemcitabine can maximize the efficacy of the immune response activated by the increased neoantigen load from the DNA-damaging agents [90,91].

Similarly, in another phase II clinical trial [NCT04753879], PD-1 blockade (pembrolizumab) is combined with a PARP inhibitor (olaparib), which blocks DNA repair, alongside a gemcitabine, nab-paclitaxel, capecitabine, cisplatin, and irinotecan (GAX-CI) regimen [92,93]. Combining PD-1 blockade and PARP inhibition alongside DNA-damaging therapeutic agents aims to significantly enhance the immune response against the increased production of neoantigens formed by PDAC. Currently, this treatment strategy is being evaluated in PDAC patients who carry germline BRCA mutations (9% of all PDAC cases). 

PD-1 blockade is also being investigated in a phase II trial [NCT04887805] in combination with the receptor tyrosine kinase (RTK) inhibitor lenvatinib for PDAC patients. Lenvatinib is a well-established multiple kinase inhibitor that has seen success across various cancers [94,95,96]. As a multiple kinase inhibitor, lenvatinib inhibits vascular endothelial growth factor receptors (VEGFR1, VEGFR2, and VEGFR3) and fibroblast growth factor receptors (FGFR1-4). By blocking these pathways, lenvatinib reduces the formation of new blood vessels that supply tumors, thereby starving the tumor of nutrients and oxygen necessary for growth and survival. Lenvatinib also targets additional RTKs involved in tumor proliferation, such as platelet-derived growth factor receptor (PDGFR), c-KIT, and RET. By inhibiting these receptors, lenvatinib can directly reduce tumor cell proliferation and induce apoptosis. Combining PD-1 blockade with lenvatinib can have promising therapeutic outcomes by halting tumor progression alongside an enhanced immune response. Only patients with unresectable disease are eligible in this trial, and the treatment strategy is primarily designed as maintenance therapy. In addition, CTLA-4, a critical immune checkpoint protein found in T cells, is being explored in clinical trials for PDAC patients. CTLA-4 interacts with its ligands (CD80/CD86) to induce T-cell exhaustion [97,98]. A novel Fc-enhanced CTLA-4 antibody, botensilimab, is currently under investigation alongside nab-paclitaxel and gemcitabine as a potent immunotherapeutic strategy [NCT05630183]. The Fc-enhanced modification of the CTLA-4 antibody increases T cell priming within the tumor microenvironment and elicits a strong response [99]. Although CTLA-4 levels are typically low in PDAC patients, prior trials targeting CTLA-4 have shown some benefit in individuals with impaired mismatch repair (MMR) systems, which are directly responsible for error correction during DNA replication [100]. Impaired MMR increases the quantities of neoantigens and thus favors cancer cells’ sensitivity to immunotherapy by enhancing the expression of inflammatory cytokines and promoting T-cell activation [101].

Currently, CTLA-4 is also under investigation in combination with NLM-001, an inhibitor of the sonic hedgehog (SHH) pathway [NCT04827953] [102]. Tumor-derived SHH activates CAFs and increases extracellular matrix deposition within the desmoplastic stroma [103,104,105]. Inhibiting the SHH pathway in PDAC disrupts tumor growth and modifies the composition of CAFs, increasing inflammatory CAFs and decreasing myofibroblastic CAFs. However, SHH pathway inhibition can also decrease cytotoxic T lymphocytes within the tumor, alongside an increase in CD4^+^ T helper cells, resulting in no net change in overall CD3^+^ T cell infiltration [106].

Expanding on novel approaches in immune checkpoint blockade for treating pancreatic cancers, a phase 2 trial [NCT02866383] has introduced a novel therapeutic strategy for treating refractory metastatic PDAC. This strategy combines stereotactic body radiation therapy (SBRT) with the checkpoint inhibitors nivolumab and ipilimumab under the study name Checkpoint Inhibition in Pancreatic Adenocarcinoma (CheckPAC). Nivolumab binds to PD-1 and blocks its interactions with PD-L1 and PD-L2, allowing T cells to remain activated and target cancer cells more effectively [107]. On the other hand, ipilimumab binds to CTLA-4 and prevents its interactions with CD80/CD86 on antigen-presenting cells, allowing for enhanced activation and function of T cells [108]. Because nivolumab and ipilimumab target different immune checkpoints, they are often combined for a synergistic effect, maximizing the immune system’s response to cancer [109,110,111]. In the CheckPAC trial, eighty-four patients received at least one dose of study treatment, which led to decreased levels of serum interleukin-6, interleukin-8, and C-reactive protein [112]. These reductions were associated with better overall survival. These findings support the continuation of the Checkpoint Inhibition and Vaccination (CheckVAC) trial, which is discussed later.

Lastly, a recent phase 2 trial [NCT05116917] investigates the potential synergy between influenza vaccination and immune checkpoint inhibitors in PDAC patients. The inspiration for this trial was the finding that influenza vaccination correlated with improved survival outcomes independent of anticancer treatment efficacy [113]. This may be due to the influenza vaccine stimulating increased T- and B-cell activation and promoting an interferon-gamma response. Therefore, it would be a promising candidate with immune checkpoint inhibitors. The overall aim of this trial is to determine the overall response rate (ORR), duration of response (DoR), disease control rate (DCR), progression-free survival (PFS), and overall survival (OS) in patients.

The clinical trials conducted so far have provided valuable insights into both the challenges and potential of immunotherapy for treating PDAC. These studies indicate that understanding the complex tumor immune microenvironment of PDAC is critical for devising novel approaches in immunotherapy to overcome the ability of PDAC to adapt. Penetrating the dense stroma of PDAC is also of therapeutic concern, as this fibrotic barrier poses a substantial barrier in the delivery and efficacy of immunotherapeutic agents. Overcoming this stromal barrier is essential for improving immune cell infiltration and function within the tumor microenvironment. In the following sections, we will explore emerging strategies that may further enhance the efficacy of immunotherapy for PDAC patients.

## 8. Potential Novel Approaches to PDAC Immunotherapy

### 8.1. Enhancing Antigen Presentation

Enhancing a tumor’s antigen-presentation machinery can make the tumor more visible to the immune system. This strategy can be achieved by upregulating MHC molecules or by introducing agents that promote the release of tumor-specific antigens. In PDAC, low MHC I expression helps tumors evade immune surveillance by avoiding recognition by cytotoxic T lymphocytes. This downregulation contributes to the immunologically “cold” tumor microenvironment typical of PDAC [114]. Methods currently being explored to increase the amount of antigen presentation and enhance immune recognition of tumors include FLT3L and CD40 agonists [115,116,117,118]. FLT3L can improve the migration of conventional dendritic cells from the bone marrow, while CD40 promotes cDC activation and increases overall MHC expression [119,120]. These targets are particularly appealing for integrating the innate and adaptive immune systems to combat cancer more effectively. One approach to targeting MHC I has been the development of mRNA vaccines that focus on MHC antigen presentation [121].

### 8.2. Cancer Vaccines

Cancer vaccines represent a promising frontier in oncology, leveraging the body’s immune system to target and eradicate cancer cells [122]. These vaccines can be designed to elicit an immune response specifically against tumor-associated antigens, providing a personalized approach to cancer treatment that avoids the complications of traditional chemo and radiotherapy (Figure 2). Various strategies have been employed in developing cancer vaccines, primarily focusing on three main approaches: DNA-based, mRNA-based, and peptide-based vaccines [123,124]. DNA vaccines utilize plasmid DNA to encode tumor antigens, inducing an immune response. mRNA vaccines, like those developed for COVID-19, involve the delivery of messenger RNA encoding cancer antigens, prompting the body to produce and present these antigens to the immune system. Peptide vaccines, conversely, directly introduce tumor-specific peptides to stimulate an immune response. Each approach offers unique advantages and challenges, which will be discussed in detail.

### 8.3. Neoantigen-Based mRNA Vaccines

mRNA vaccines are emerging as prominent candidates for the precision treatment of PDAC [125,126,127,128,129,130]. mRNA vaccines generate robust anti-tumor responses that engage innate and adaptive immune systems. Initially, the innate immune system recognizes foreign mRNA via pattern recognition receptors on antigen-presenting cells like dendritic cells [131]. This detection, in turn, triggers a cascade of pro-inflammatory signaling pathways that enhance innate immune function. mRNA vaccines can also stimulate adaptive immunity by facilitating the processing of non-self mRNA-encoded proteins into peptides, which are then presented on MHC-I and transported to the cell surface, where they activate CD8^+^ T cells.

Additionally, these neoantigens can be directed through the Golgi bodies to endosomes to engage in the MHC-II presentation pathway to activate CD4^+^ T cells. The mRNA vaccine response is amplified by the upregulation of co-stimulatory molecules (such as CD40 and CD86) on antigen-presenting cells, which enhances antigen presentation and T-cell activation. Activated antigen-presenting cells, including macrophages and dendritic cells, also present antigens to B cells, initiating an antibody response. Recent clinical trials involving an mRNA-based vaccine for pancreatic cancer have shown promising results, particularly in a study conducted at Memorial Sloan Kettering Cancer Center [132]. The vaccine, known as autogene cevumeran (RO7198457), includes an individualized mRNA neoantigen vaccine containing up to 20 neoantigens identified in each patient’s tumor. It led to durable and functional T-cell responses in patients with resectable pancreatic cancer and was associated with a reduced risk of disease recurrence. In this phase I trial [NCT04161755], 16 patients received R07198457 combined with the checkpoint inhibitor atezolizumab and a chemotherapy regimen. Half the participants developed robust immune responses against one or more tumor neoantigens. Importantly, these T cells were long-lasting and maintained their ability to respond to neoantigens for up to three years post-vaccination. Such findings underscore the potential of a vaccine to induce a robust immune response and contribute to delayed disease recurrence in pancreatic cancer.

Safety is paramount in precision therapy, and mRNA vaccines exhibit several key safety advantages over other vaccine platforms. For example, the production and delivery of mRNA vaccines do not involve toxic chemicals, reducing potential harm to manufacturing personnel and patients [133,134]. Additionally, mRNA vaccine production mitigates the risk of contamination with adventitious viruses that can be introduced during the culture of host cells, a concern associated with other vaccine platforms like viral vectors, inactivated viruses, live viruses, and subunit protein vaccines. The rapid manufacture of mRNA vaccines also reduces the window of opportunity for contaminating microorganisms during production. Unlike other therapeutic modalities, mRNA cannot integrate into the host genome. Moreover, the adjustable half-life of mRNA allows for precise control over the duration and intensity of protein expression. This approach enhances safety by allowing for the modulation of immune responses and potential side effects.

Nonetheless, some challenges need to be addressed with mRNA vaccines. The inherent properties of naked mRNA, such as its size, degradability, and charge, can impede efficient cellular uptake and cytoplasmic entry, except in cases like immature dendritic cells that can efficiently internalize mRNA via the macropinocytosis pathway [135]. To enhance the effective delivery of mRNA into antigen-presenting cells, appropriate mRNA formulations (e.g., liposomes, polyplexes, polysomes, and lipoplexes) and administration routes must be judiciously selected and optimized. Once successful mRNA delivery is achieved, the in vivo half-life of transcribed mRNA requires careful regulation, as various factors influence the pharmacodynamic and pharmacokinetic properties of mRNA-based therapeutics. Structural improvements to mRNA, such as optimizing poly(A) tails, 5′ cap structures, and untranslated regions, are vital for enhancing mRNA stability and overall durability. In addition to delivery and stability considerations, immunogenicity must be a focal point in mRNA vaccine design. Emerging evidence suggests a complex interplay between mRNA and its associated immune response. For example, exogenous RNA stimulates the production of type I interferon through innate immunity pathways, but excessive production can promote the degradation of both ribosomal RNA and cellular mRNA [136]. Strategies to mitigate immunogenicity include sequence optimization and post-transcriptional purification, which can reduce innate immune responses while preserving mRNA translation [137].

Furthermore, enhancing the immunostimulatory properties of mRNA by incorporating adjuvants, such as TriMix (mRNA encoding CD70, CD40L, and TLR4), can augment the potency of cancer mRNA vaccines [138]. TriMix, for instance, enhances the immunogenicity of unmodified naked mRNA, facilitating the cytotoxicity of T lymphocytes and the maturation of dendritic cells [139]. These advancements in mRNA vaccine construction are essential for improving their efficacy in treating pancreatic cancer. Furthermore, targeting KRAS mutations, one of the most commonly present mutations in PDAC, through immunotherapies has been especially challenging. A trial [NCT03948763] utilizing mRNA-5671 (V941), which is a tetravalent vaccine targeting KRAS G12D, G12V, G13D, and G12C (Moderna Inc.), was eventually discontinued in 2022 as it did not meet efficacy endpoints and, consequently, no further progress or updates on V941 have been announced. This issue underscores the necessity for improved vaccine platforms and combinatorial therapies that can levy the immunological response provoked to infiltrate and target antigen-expressing cancer cells.

As mentioned in the CheckPAC trial, some patients experienced improved clinical response rates associated with lowered TGF-β levels, which led to the initiation of the CheckVAC trial [NCT05721846]. CheckVAC is currently exploring the combination of a TGFβ-15 peptide vaccine with nivolumab and ipilimumab treatment. TGFβ-15 is a formulated peptide vaccine containing a TGFβ-derived peptide alongside Montanide ISA-51 as an adjuvant [140]. Upon administration, TGFβ-15 aims to restore and enhance an immunological anti-tumor response by stimulating the host immune system to mount a cytotoxic T-lymphocyte response against TGFβ-expressing immunosuppressive cells in the TME, including TAMs, MDSCs, DCs, Tregs, and CAFs.

Another cancer vaccine that is currently being tested is the combination of TG01 Vaccine/QS-21 Stimulon with or without immune checkpoint inhibitor balstilimab as maintenance therapy following adjuvant chemotherapy in patients with resected pancreatic cancer (TESLA) trial [NCT05638698]. TG01 is an experimental vaccine designed to provoke an immune response against cancer cells by targeting the seven most prevalent codon 12 and 13 oncogenic mutations in KRAS with synthetic RAS peptides. QS-21, derived from the soap bark tree, is a vaccine adjuvant known for stimulating both humoral and cell-mediated immunity, further boosting the immune response induced by TG01.

A new cancer vaccine clinical trial [NCT05964361] focuses on enhancing the body’s immune response by targeting the Wilms tumor 1 (WT1) protein. WT1 is a transcription factor that plays a crucial role in both normal development and tumorigenesis [141]. WT1 is highly overexpressed in various malignancies, including pancreatic cancer [142]. WT1 supports tumor progression by promoting cell proliferation, inhibiting apoptosis, and enhancing angiogenesis. This trial investigates the feasibility of developing vaccines specifically targeting the Wilms’ Tumor-1 (WT1) antigen alongside a novel IL15-trans presentation mechanism on their cell surface. IL15, known for its role in supporting natural killer cell function, promoting T cell memory formation, and enhancing immune response, is expected to enhance the immunogenicity of dendritic cells towards WT1-expressing cancer cells [143,144].

### 8.4. Neoantigen-Based Peptide Vaccines

In addition to mRNA vaccines, peptide vaccines are another emerging alternative for PDAC patients. Peptide vaccines are designed to induce the host’s immune system to target neoantigens, which are unique to tumor cells due to mutations and are absent in normal cells. The production of neoantigen-based peptide vaccines typically involves several steps. Initially, tumor samples are obtained from the patient, and the genetic material of the tumor cells is sequenced to identify neoantigens produced by the cancer cells. Bioinformatics tools are then employed to predict which neoantigens are most likely to be recognized by the patient’s immune system. Once suitable neoantigens are identified, corresponding synthetic peptides are manufactured. These peptides are then formulated into a vaccine, often with an adjuvant to enhance the immune response. The personalized vaccine is subsequently administered to the patient. This process tailors the treatment to the unique genetic profile of each patient’s cancer. It also minimizes damage to healthy tissue akin to other vaccination methods previously discussed—a common side effect of broad-based chemotherapeutic and radiotherapeutic treatments.

An example of the successful development of a personalized neoantigen-based peptide vaccine is iNeo-Vac-P01 for patients with advanced PDAC who had exhausted standard treatment options [145]. Notably, the vaccine was well-tolerated, recording no severe adverse events. The median overall survival (OS) and progression-free survival (PFS) for this cohort surpassed those of prior clinical studies, suggesting the vaccine’s potential to extend life for patients with this aggressive form of cancer. Some participants also received the neoantigen-based peptide vaccine with chemotherapy or immune checkpoint inhibitors to boost clinical outcomes. Central to the vaccine’s success was its ability to stimulate T-cell solid responses against specific tumor antigens. This study identified induced T-cell reaction alterations in peripheral blood T-cell subsets and reported that higher interferon-gamma levels were detected in patients with better OS. This research also signifies strategic timing for vaccine administration, suggesting that minimized tumor burden could optimize immune infiltration. This study proposed combining the neoantigen-based peptide vaccine with anti-CTLA-4 antibodies, presenting a potent therapeutic combination for further evaluation.

Another peptide-based vaccine strategy for pancreatic cancer involves harnessing neoantigens’ specificity to stimulate an immune response against cancer cells. One example is SLP vaccines, which consist of synthetic peptides that mimic neoantigens found in cancer cells [146]. They are typically longer than traditional short peptides and can encompass more of the neoantigen sequence. When these SLP vaccines are administered, dendritic cells capture the synthetic peptides, process them, and present them on their cell surface along with MHC molecules. T cells become activated when they encounter the dendritic cells presenting the synthetic peptides. The activated T cells undergo clonal expansion, resulting in a population of T cells primed to target the tumor. SLP vaccines have also been demonstrated to stimulate the formation of memory T cells, leading to long-term immune surveillance. A new phase 1 trial [NCT05013216] is set to evaluate the safety and immunogenicity of a pooled mutant-KRAS (mKRAS) SLP peptide vaccine combined with the poly-ICLC adjuvant in 20 patients at high risk of developing PDAC. These high-risk groups were identified as individuals who 1) have familial pancreatic cancer relatives or 2) are germline mutation carriers with an estimated lifetime risk of pancreatic cancer of ~10% or higher. The vaccine formulation includes synthetic long peptides representing six common mKRAS subtypes: G12D, G12R, G12V, G12A, G12C, and G13D. These peptides are combined with the poly-ICLC adjuvant to enhance the vaccine’s effectiveness.

A promising SLP immunotherapy platform has recently been underway involving ELI-002. ELI-002 is composed of a lipid-conjugated immune-stimulatory oligonucleotide (Amph-CpG-7909) and a combination of lipid-conjugated peptide-based antigens (Amph-Peptides) designated by “NP” to target a broad spectrum of KRAS mutations expressed in various cancers. The first trial [NCT04853017] involving ELI-002 was conducted with ELI-002 2P, which codes for Amph-modified KRAS peptides: Amph-G12D and Amph-G12R. The dose-escalation study was designed to assess the safety and efficacy of SLP ELI-002 as an immunotherapy that could be an adjuvant treatment for patients with KRAS/NRAS viral oncogene homolog-mutated PDAC, among other solid tumors, and has since been transitioned into another trial [NCT04853017] utilizing the ELI-002 7P formulation [147]. ELI-002 7P incorporates seven different Amph-Peptides: G12D, G12R, G12V, G12A, G12C, G12S, and G13D. This trial currently assesses 156 subjects across three planned dose levels, with the potential for additional cohorts based on safety reviews and preliminary pharmacodynamic responses. The primary objective is to establish a recommended Phase 2 dose and set the maximum tolerated dose.

Another SLP immunotherapy trial [NCT04117087] underway is a single-arm, open-label, first-in-human phase I study of a pooled SLP vaccine targeting six mKRAS subtypes: G12D, G12R, G12V, G12A, G12C, and G13D. Treatment is combined with the checkpoint inhibitors ipilimumab and nivolumab in patients with resected PDAC. This study aims to evaluate the fold change in interferon-producing mutant-KRAS-specific CD4/CD8^+^ T cells after vaccination alongside any drug-related adverse events. Secondary measures are determining the overall DFS, ORR, OS, and PFS in these treated patients.

Lastly, a phase I clinical trial [NCT05846516] is evaluating the experimental immunotherapy platform KISIMA-02 for pancreatic cancer patients. The KISIMA platform is a vaccine that is comprised of three components: a cell-penetrating peptide (CPP) for the transportation of the vaccine contents across the cell membrane, antigens that can be tailored to each indication, and a TLR peptide agonist that acts as an adjuvant [148]. KISIMA-02 is a combination treatment for patients with KRAS G12D/G12V-mutated PDAC. This regimen includes the KISIMA therapeutic protein vaccine (ATP150 or ATP152), an oncolytic viral vector (VSV-GP154), and the immune checkpoint inhibitor ezabenlimab. This study is designed to assess the safety and tolerability of the KISIMA-02 regimen before examining its effect on delaying tumor recurrence.

### 8.5. Neoantigen-Based DNA Vaccines

While many trials have focused on inducing robust CD4 and CD8 T cell responses through peptide and mRNA vaccines, another open-label phase I trial [NCT03122106] aimed to study and evaluate the safety and immunogenicity of a neoantigen DNA vaccine strategy in PDAC patients following surgical resection and adjuvant chemotherapy [149]. The neoantigen DNA vaccine incorporates prioritized neoantigens and personalized mesothelin epitopes and is administered via electroporation. Results indicated that the neoantigen DNA vaccine could stimulate immune responses, as evidenced by the presence of these T cells in patients’ blood samples after vaccination. However, challenges such as variability in neoantigen identification and vaccine formulation were noted, highlighting the complexity of developing personalized DNA-based cancer vaccines.

### 8.6. Re-Engineered Pancreatic Cancer Cells as a Vaccine Modality

The pancreatic GVAX platform is another example of a novel strategy that has seen success in other cancers, with the pancreatic version being a vaccine composed of genetically modified pancreatic cancer cells [150,151]. These engineered pancreatic cells secrete granulocyte-macrophage colony-stimulating factor (GM-CSF) to facilitate a strong dendritic cell-dependent immunological response. GVAX induces the formation of tertiary lymphoid structures within the tumor, providing a localized site for immune cell activation for an effective immune response. In certain trial arms, GVAX is combined with immune checkpoint inhibitors like nivolumab and ipilimumab. Combining GVAX with these checkpoint inhibitors aims to prime the immune system against the tumor and prevent immune escape. GVAX is also commonly administered with low-dose cyclophosphamide, which selectively depletes Tregs that suppress immune responses, thus enhancing the vaccine’s effectiveness [152]. A three-arm, phase 2 clinical trial [NCT02451982] is currently underway, aiming to determine the effect of co-treatment with GVAX and CY on patient outcomes. Another GVAX-related phase 2 clinical trial [NCT03190265] is focused on the investigation of ORR and AEs with the combination therapy of CRS-207, nivolumab, and ipilimumab, with or without the GVAX (and CY). CRS-207 is a live-attenuated, double-deleted Listeria monocytogenes strain engineered to express the pancreatic tumor-associated antigen mesothelin [153,154]. By infecting antigen-presenting cells, CRS-207 helps break immune tolerance by inducing a potent innate and adaptive immune response, including activating T cells specific to mesothelin. Overall, GVAX provides a broad array of tumor antigens in the context of GM-CSF. At the same time, CRS-207 focuses on a specific antigen, mesothelin, enhancing the overall presentation of tumor antigens to the immune system. This combination can lead to a more robust activation and expansion of tumor-specific T cells, as both therapies independently activate and mature dendritic cells. CRS-207’s ability to break immune tolerance complements GVAX’s ability to stimulate an immune response, which may lead to a more effective and sustained anti-tumor immune response. Targeting multiple tumor-associated antigens between GVAX and CRS-207 would also reduce the likelihood of immune escape by tumor cells.

### 8.7. Combination of Epigenetic Drugs with Immune Checkpoint Inhibitors

Targeting the epigenetic modifiers of pancreatic cancer represents another promising approach that can offer new therapeutic avenues for this malignancy, which is notoriously resistant to conventional treatments. Epigenetic modifications, which involve reversible changes in gene expression without altering the underlying DNA sequence, play a pivotal role in the development and progression of pancreatic cancer [155]. These modifications include DNA methylation, histone modification, and chromatin remodeling, among others, all of which can contribute to the dysregulation of gene expression, leading to tumor growth, metastasis, and resistance to treatment. Drugs such as azacitidine, decitabine, and tazemetostat, which target epigenetic processes, aim to reverse these aberrant modifications, restoring normal gene expression and enhancing the effectiveness of other treatments, including immunotherapies. Azacitidine and decitabine are both DNA methyltransferase inhibitors (DNMTi) that work by inhibiting DNA methylation, a common epigenetic modification in cancer cells. In pancreatic cancer, hypermethylation can silence genes responsible for apoptosis, DNA repair, and immune response. By demethylating DNA, azacitidine and decitabine can reactivate these crucial genes, thereby re-sensitizing cancer cells to chemotherapy and immune checkpoint inhibitors [156,157]. This demethylation process also leads to the expression of cancer-testis antigens and other tumor-associated antigens, making the cancer cells more recognizable to the immune system [158,159]. On the other hand, histone deacetylase inhibitors (HDACi) inhibit HDACs, which remove acetyl groups from histone tails, resulting in a more relaxed chromatin structure. This increased chromatin accessibility can reactivate tumor suppressor genes that were previously silenced in pancreatic cancer cells, leading to the inhibition of tumor growth and induction of apoptosis.

The combination of epigenetic drugs with ICIs could be a promising strategy in the treatment of pancreatic cancer. For example, epigenetic therapies can remodel the tumor microenvironment by altering the expression of immune-related genes, thereby enhancing the infiltration and activity of immune cells within the tumor. For instance, the demethylating effects of drugs like azacitidine and decitabine can increase the expression of PD-L1 and other immune checkpoints, making cancer cells more susceptible to ICIs [160]. Additionally, these drugs can upregulate genes associated with antigen presentation machinery and other immune-related pathways, effectively transforming immunologically “cold“ tumors into “hot“ tumors, which are more responsive to immunotherapy. A phase I/II clinical trial [NCT04257448] is currently underway to assess the efficacy of combining azacitidine, romidepsin, lenalidomide, and durvalumab alongside standard of care in pancreatic cancer patients. Romidepsin is a histone deacetylase inhibitor (HDACi), lenalidomide is an immunomodulatory drug that alters cytokine production and regulates T cell co-stimulation, and durvalumab is a PD-L1 inhibitor. In addition, a phase Ib/II clinical trial [NCT06454448] is underway, which combines decitabine with adebrelimab, a PD-L1 inhibitor.

Tazemetostat, on the other hand, is an enhancer of zeste homolog 2 (EZH2) inhibitor. EZH2 is a histone methyltransferase that plays a key role in the polycomb repressive complex 2 (PRC2) [161,162]. EZH2 is often overexpressed in pancreatic cancer, leading to the silencing of tumor suppressor genes through histone methylation. By inhibiting EZH2, tazemetostat reduces histone methylation, reactivating these suppressed genes and thereby inhibiting tumor growth. Moreover, EZH2 inhibition can enhance the expression of neoantigens, further promoting an anti-tumor immune response [163]. A phase II clinical trial [NCT04705818] assessing tazemetostat’s efficacy combined with durvalumab, a PD-L1 inhibitor, is currently underway.

Recent clinical trials combining epigenetic drugs with ICIs in pancreatic cancer have shown encouraging results, indicating that this combination can be advantageous for overcoming the immune resistance characteristic of pancreatic cancer. While the epigenetic drug class has shown promising effects in preclinical studies and early-phase trials, there are still critical side effects to consider. The most common dose-limiting side effects of epigenetic drugs reported include fatigue, gastrointestinal symptoms (nausea, vomiting, and diarrhea), thrombocytopenia, neutropenia, anemia, and liver toxicity.

## 9. Discussion

There is an urgent need to innovate and improve therapeutic strategies for treating pancreatic cancer, as it is predicted to become the second leading cause of cancer-related death by 2030. Current treatment protocols heavily rely on chemotherapeutic agents such as gemcitabine and FOLFIRINOX; however, their efficacy is significantly compromised by high toxicity and the development of chemoresistance. Despite advancements in cancer therapeutics such as immunotherapy, PDAC remains a dauntingly complex cancer to treat due to its distinctive structural composition. The dense fibrotic stroma characteristic of PDAC creates a physical barrier that blocks immune cell infiltration. In addition, PDAC’s TME poses a unique challenge as it is replete with immunosuppressive elements, including Tregs, MDSCs, and TAMs, which contribute to a robust barrier against an effective immune response. Recent efforts are focused on enhancing the delivery and efficacy of immunotherapy by dissecting the complexities of the PDAC TME. Novel strategies aim to target the dense desmoplastic stroma, neutralize the effects of immunosuppressive cytokines, and enhance the presentation of tumor antigens to induce a more robust immune response. For example, personalized neoantigen vaccines and precision targeting of cancer cells have shown considerable promise in enhancing the immune response directly at the tumor site, thus improving the specificity and effectiveness of treatment. Emerging approaches also involve the utility of dual-targeting strategies, where immune checkpoint blockade is used with agents that modulate the immune environment. Approaches focused on targeting specific tumor antigens to enhance the immunogenicity of the tumors have also gained attention. Furthermore, developing targeted therapies such as the TG01 vaccine, specifically addressing KRAS mutations common in pancreatic cancer, indicates a shift towards genetic precision in immunotherapy. The direct targeting of the genetic abnormalities driving cancer progression promises a more tailored and potentially effective treatment regimen.

In summary, the promising trajectory of immunotherapy in PDAC underscores a shift towards more personalized and precisely targeted treatment regimens. Continued research and clinical trials will be crucial in moving these innovative strategies from the bench to the clinic. Integrating multidisciplinary research covering molecular biology, pharmacology, and immunology, along with robust patient engagement, will further refine and enhance the therapeutic landscape of PDAC, offering hope for improved survival and quality of life for patients with this challenging disease.

## Figures and Tables

**Figure 1 biomedicines-12-01979-f001:**
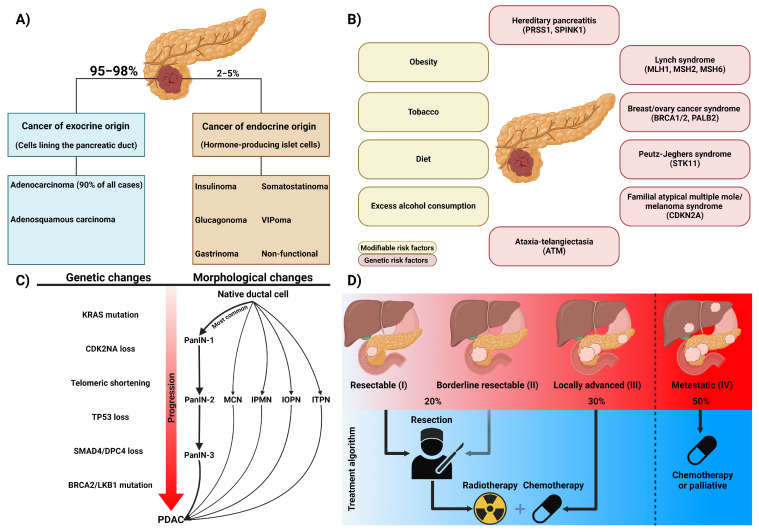
Summary of pancreatic cancer clinical features and presentation. (**A**) Distribution of pancreatic cancer histopathological cells of origin and subtype. (**B**) Known modifiable and genetic risk factors contributing to an increased risk of pancreatic cancer. (**C**) Genetic and morphological progression of native ductal cells in the pancreas towards PDAC. (**D**) Diagram of the percentage of patients diagnosed at different stages of PDAC and a generalized treatment algorithm for each stage. Abbreviations: pancreatic intraepithelial neoplasia (PanIN), mucinous cystic neoplasm (MCN), intraductal papillary mucinous neoplasm (IPMN), intraductal oncocytic papillary neoplasm (IOPN), and pancreatic intraductal tubulopapillary neoplasm (ITPN). Created with BioRender.com.

**Figure 2 biomedicines-12-01979-f002:**
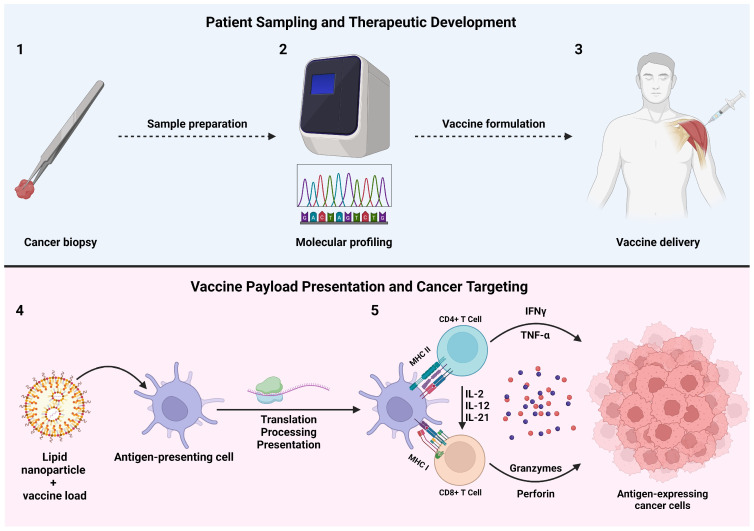
Schematic summary and mechanistic role of cancer vaccine therapeutic development and administration. Created with BioRender.com.

**Table 1 biomedicines-12-01979-t001:** Summary of clinical trials evaluating novel agents targeting PDAC tumor microenvironment (Retrieved from ClinicalTrials.gov).

Trial Identifier	Phase	Agent(s)	Molecular Targets	Mechanism	Population
**NCT03948763**	I	mRNA-5671/V941	KRASG12D KRASG12V KRASG13D KRASG12C	mRNA against KRAS elicits a T cell response	KRAS-mutated Non-small cell lung cancer, colorectal cancer or PDAC
**NCT05726864**	II	ELI-002	KRASG12D KRASG12R	Lipid-conjugated immune-stimulatory oligonucleotide and a mixture of lipid-conjugated peptide-based antigens	KRAS-mutated PDAC, colorectal cancer, NSCLC, ovarian cancer, cholangiocarcinoma bile duct cancer or gallbladder carcinoma
**NCT05013216**	I	KRAS peptide vaccine + poly-ICLC	KRASG12D KRASG12R KRASG12V KRASG12 KRASG12C KRASG13D	Targeted long peptide vaccine against mutant KRAS	High risk of PDAC by family history or germline mutation status
**NCT04117087**	I	KRAS peptide vaccine plus poly-ICLC	KRASG12C KRASG12V KRASG12D KRASG12A KRASG13D KRASG12R	Targeted long peptide vaccine against mutant KRAS with the addition of enhanced cell-mediated immune response	Resected PDAC after neoadjuvant and/or adjuvant chemotherapy and/or radiation
**NCT03122106**	I	Neoantigen peptide vaccine plus poly-ICLC	Prioritized neoantigens and personalized mesothelin epitopes	Neoantigen peptide vaccine will be capable of generating neoantigen-specific CD4+ and CD8^+^ T cell responses	PDAC following surgical resection and adjuvant chemotherapy
**NCT04161755**	I	RO7198457 (mRNA-based personalized tumor vaccine)	Tumor-associated antigens	APCs take up the mRNA-based vaccine and express tumor-associated antigens, leading to cytotoxic and memory T cell immune responses against the tumor-associated antigens	PDAC, undergoing curative intent resection
**NCT02600949**	I	Synthetic personalized tumor-associated peptide vaccine therapy	Tumor-associated antigens	APC take up mRNA-based vaccine and express tumor-associated antigens, leading to cytotoxic and memory T cell immune responses against the tumor-associated antigens	Advanced PDAC and colorectal cancer
**NCT04683939**	I/IIA	BNT141	Claudin 18.2	APC take up the mRNA-based vaccine and express tumor-associated antigens, resulting in cytotoxic and memory T cell immune responses against the tumor-associated antigens.	Patients With CLDN18.2-positive Solid Tumors including pancreatic cancers.
**NCT04999969**	II	AZD0171	LIF	AZD0171 is a humanized monoclonal antibody that binds with high affinity to LIF, promoting antitumor inflammation by modulating TAMs and inhibiting cancer stem cells, thereby slowing tumor growth.	First line (1L) metastatic PDAC
**NCT04580485**	I	INCB106385	A2A and A2B receptors	Direct A2A/A2B antagonist. ATP released by dying cells is converted into adenosine, a potent suppressor of immune cell activity. Blocking both adenosine receptors on immune cells can be used as an immunotherapy to enhance anti-tumor immune responses.	CD8 T-cell-positive advanced solid tumors including PDAC
**NCT04989387**	I	INCA00186	CD73	Humanized monoclonal antibody antagonist of CD73, blocking the production of adenosine, thereby restoring immune function by reducing adenosine levels.	Advanced solid tumors; squamous cell carcinoma of the head and neck (SCCHN) and specified gastrointestinal (GI) malignancies.
**NCT05688215**	I/II	Zimberelimab and quemliclustat	Zimberelimab: PD-1 quemliclustat: CD73	Selective small molecule inhibitor of CD73, blocking the production of adenosine, thereby restoring immune function by reducing adenosine levels.	Borderline resectable PDAC or locally advanced.
**NCT05632328**	II	AGEN1423 and Balstilimab	AGEN1423: CD73 and TGFβ, Balstilimab: PD-1	1) Preferential localization within the tumor microenvironment (TME) due to its CD73 targeting moiety; 2) Ability to reduce adenosine concentration in the TME by inhibiting CD73 enzymatic activity; and 3) Inhibition of the immunosuppressive effect of TGFβ.	Advanced PDAC
**NCT05562297**	II	Neoadjuvant/Adjuvant Sintilimab, Nab-paclitaxel, and Gemcitabine	Sintilimab: investigational PD-1 inhibitor	Binds to PD-1 molecules on the surface of T-cells, blocks the PD-1/PD-Ligand 1 (PD-L1) pathway, and reactivates T-cells to kill cancer cells	Resectable/Borderline Resectable PDAC
**NCT04753879**	II	GAX-CI (gemcitabine, nab-paclitaxel, capecitabine, cisplatin, and irinotecan)	Gemcitabine: nucleoside analog; Nab-paclitaxel: microtubule inhibitor; Capecitabine: thymidylate synthase inhibitor; Cisplatin: DNA crosslinks; Irinotecan: topoisomerase I inhibitor.	Gemcitabine incorporates into DNA and inhibits ribonucleotide reductase. Nab-paclitaxel stabilizes microtubules, inducing apoptosis. Capecitabine, metabolized into 5-fluorouracil, inhibits thymidylate synthase, impairing DNA synthesis. Cisplatin forms DNA cross-links, preventing replication and transcription, leading to cell death. Irinotecan inhibits topoisomerase I, causing DNA damage during replication.	Metastatic PDAC
**NCT04887805**	II	Lenvatinib and Pembrolizumab	Lenvatinib: VEGFR FGFR, *RET*, PDGFRα, and *KIT* Pembrolizumab: PD-1	Inhibition of VEGF receptors prevent tumor angiogenesis, while inhibiting FGFR, RET, PDGFRα, and KIT halts the further proliferation of malignant cells.	Advanced Unresectable PDAC
**NCT05630183**	II	Botensilimab	Fc-enhanced multifunctional anti-cytotoxic T-lymphocyte-associated protein 4 (CTLA-4)	Expand therapy to cold or poorly immunogenic solid tumors.	Metastatic PDAC
**NCT04827953**	IB/IIA	NLM-001	Hedgehog Inhibitor	NLM-001 targets the Hh pathway and disrupts the tumor microenvironment (TME) by decreasing cancer-associated fibroblasts and promoting immune cell infiltration into the TME.	Advanced PDAC
**NCT02866383**	II	Nivolumab & Ipilimumab & Radiotherapy	Nivolumab: PD-1, Ipilimumab: CTLA-4	Nivolumab blocks PD-1, preventing T cell inhibition and enhancing the immune response against tumor cells. Ipilimumab blocks CTLA-4, further promoting T cell activation and proliferation. Radiotherapy can cause immunogenic cell death, leading to the release of tumor antigens.	metastatic PC or BTC refractory or intolerant to at least one line of prior systemic chemotherapy with gemcitabine or platinum-containing regimens.
**NCT05721846**	I	TGFβ-15 Peptide Vaccine	TGFβ1-derived peptides (MHC-I and MHC-II-restricted)	Immunosuppression and fibrosis are driven by TGFβ. Targeting the immunosuppressive and fibrotic TME in PDAC leads to increased infiltration of CD8^+^ T cells and a higher intratumoral M1/M2 macrophage ratio.	Refractory PDAC
**NCT05116917**	II	Influenza Vaccine	N/A	The influenza vaccine promotes the activation of T cells and B cells, triggering an interferon-gamma response.	Pancreatic Cancer
**NCT05638698**	II	Tg01 Vaccine/Qs-21 Stimulon™ (PD-1)	RAS-neoantigen peptide vaccine targeting the seven most frequent codon 12-13 RAS mutations	Activates mutant RAS-specific CD4^+^ and CD8^+^ T-cell responses.	Surgically resected Stage 1-3 RAS mutant PDAC who have ctDNA in the blood despite prior therapy.
**NCT05964361**	I/II	Interleukin-15-transpresenting Wilms’ Tumor Protein 1 Autologous Dendritic Cell Vaccination	WT1	By targeting the tumor-associated antigen Wilms’ tumor-1 (WT1), the tumor-antigen specificity stimulates antigen-specific T cells using PD-L-silenced IL-15 dendritic cells (DCs).	Esophageal Cancer Pancreas Cancer Ovarian Cancer Liver Cancer.
**NCT05846516**	I	VSV-GP154, ATP150, ATP152, Ezabenlimab	KRAS G12D, KRAS G12V	Upon administration of ATP150 and ATP152, the Z12 moiety targets, binds to, and penetrates antigen-presenting cells (APCs), specifically dendritic cells (DCs). This promotes the loading of epitopes into the DCs and transports antigenic cargo into both endosomal and cytosolic compartments.	KRAS G12D/G12V Mutated PDAC
**NCT02451982**	II	GVAX + PD-1 inhibitors	Antigen-presenting tumor cells	GVAX vaccination stimulates local production of granulocyte-macrophage colony-stimulating factor (GM-CSF) at the vaccine site, leading to a systemic tumor-specific immune response.	Resectable PDAC
**NCT03190265**	II	GVAX Pancreas Vaccine (With Cyclophosphamide) + nivolumab and ipilimumab	Antigen-presenting tumor cells	GVAX enhances the presentation of tumor antigens to the immune system. Cyclophosphamide further amplifies this effect by reducing regulatory T cells. Nivolumab and ipilimumab inhibit the PD-1 and CTLA-4 pathways, which would normally suppress the immune response.	Metastatic PDAC
**NCT04853017**	I	ELI-002	Lymph-node targeted amphiphile (AMP)-modified G12D and G12R mutant KRAS peptides together with an AMP-modified CpG oligonucleotide adjuvant.	Increased cytotoxic KRAS-specific T cells	KRAS Mutated PDAC
**NCT04257448**	I/II	Romidepsin Azacitidine + nab-Paclitaxel/Gemcitabine ± Lenalidomide + Durvalumab	Romidepsin: Histone Deacetylase (HDAC) Inhibitor, Azacitidine: DNA Methyltransferase (DNMT) Inhibitor, Nab-Paclitaxel: microtubules, Durvalumab: PD-L1 inhibitor	Romidepsin targets HDACs to induce apoptosis. Azacitidine inhibits DNMTs to reactivate tumor suppressor genes and induce differentiation. Nab-Paclitaxel stabilizes microtubules. Gemcitabine is incorporated into DNA to prevent synthesis. In the consolidation phase, Durvalumab enhances immune response by preventing T-cell inhibition, and Lenalidomide boosts immune activity and modulates the TME.	Advanced PDAC
**NCT04705818**	II	Tazemetostat + Durvalumab	Tazemetostat: EZH2 inhibitor, Durvalumab: PD-L1 inhibitor	Durvalumab blocks PD-L1 to enhance the immune response by preventing T-cell inhibition, while Tazemetostat inhibits EZH2, a histone methyltransferase that represses gene expression.	Advanced PDAC
**NCT06454448**	Ib/II	Adebrelimab, Decitabine, paclitaxel, gemcitabine	Adebrelimab: Anti-PD-L1, Decitabine: DNA Methyltransferase Inhibitor, Paclitaxel: microtubules, Gemcitabine: nucleoside analog	Adebrelimab blocks the PD-1 checkpoint receptor on T-cells, enhancing the immune response against PDAC. Decitabine induces hypomethylation of DNA, which can reactivate tumor suppressor genes, promoting apoptosis. Paclitaxel stabilizes microtubules, disrupting the mitotic spindle, leading to cell death. Gemcitabine incorporates into DNA, inhibiting DNA synthesis and causing DNA damage.	Metastatic PDAC

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
