# Peer review of "The Road Ahead in Pancreatic Cancer: Emerging Trends and Therapeutic Prospects"

_biomedicines, 2024, doi:10.3390/biomedicines12091979_

Round 1
Reviewer 1 Report
Comments and Suggestions for Authors
It is a well-organized review about the topic of pancreatic cancer therapeutics. The summary of all effective theraputic approaches is very necessary to stimultate the discussion about how to treat pancreatic cancers. Overal, this paper is a comprehensive description of diverse methods to combat pancreatic cancer, especially focusing on the TME targeting. For the higher readability, I have to refer some necessary improvements to be noted befor the final acceptance for publication.
1. Some paragraphs are too long in the same subtitle. Maybe a further subdivision is highly recommended for cancer vaccine.
2. Without any figures, the readers can not easily get the useful information. The illustration of pancreatic cancer formation, the current theraptuics of PDAC, the example of mRNA vaccine in PDAC treatment will be more welcome.
Reviewer 2 Report
Comments and Suggestions for Authors
The topic “pancreatic cancer” is interesting, however, the manuscript is too short to cover new aspects of diagnosis or treatment. Moreover, I would like to know this work's novelty since many comprehensive review papers have been published.
Here are my specific comments:
1) The abstract is extremely short and does not have any results/findings from previous research. Usually, the abstract should consist of 1-2 sentences of background, one sentence of the aims of the review, and 4-5 sentences of the most important results must be mentioned. The numerical results are preferred. The abstract should be rewritten.
2) I strongly recommend that the authors draw a graphical abstract at the end of the introduction.
3) I recommend also making numbers for the sections and subsections.
4) The tumor microenvironment of PDAC could be explained more, including the role of stiffness of the microenvironment in the regulation of tumor invasion.
5) The authors did not explain the diagnosis especially the early diagnosis of pancreatic cancer. Although PDAC is not the first or second killer cancer, however, its five-year survival within affected patients is very low (below 5%). Early diagnosis could help patients not to be diagnosed at the last stages. However, the authors did not discuss it at all.
6) The authors did not discuss epi-drugs to fix epigenetic aberrations or combination therapy (Immunotherapy +chemotherapy or immunotherapy + epi-drug).
7) Table 1 does not have a caption!!! The font size is too small and cannot be seen. It has to be revised to make it more visible.
Round 2
Reviewer 1 Report
Comments and Suggestions for Authors
All my concerns have been well addressed. This paper can be accepted for publication in its current form.
Reviewer 2 Report
Comments and Suggestions for Authors
The authors improved the quality of the work. However, it could expand more to cover all aspects of the work but it can be accepted at this stage.